# Walking Speed Modulates Neck–Shoulder Strain During Smartphone Use with Backpack Load

**DOI:** 10.3390/healthcare13233141

**Published:** 2025-12-02

**Authors:** Yi-Lang Chen, Dinh-Dung Nguyen

**Affiliations:** Department of Industrial Engineering and Management, Ming Chi University of Technology, New Taipei 243303, Taiwan; m12218031@mail2.mcut.edu.tw

**Keywords:** smartphone use, walking speed, backpack load, musculoskeletal strain, posture, electromyography (EMG)

## Abstract

**Background/Objectives:** The concurrent use of smartphones and backpacks presents notable ergonomic challenges for students and young adults. However, the influence of walking speed on this dual-task biomechanical strain remains unclear. This study investigated how walking speed, backpack load, and gender affect postural alignment and neck–shoulder muscle activity during smartphone use. **Methods:** Thirty healthy university students (15 males and 15 females) were assessed using a motion capture system and surface electromyography to quantify postural alignment and muscle activity. Each participant completed twelve randomized conditions comprising three backpack loads (0%, 5%, and 10% of body weight) combined with four locomotor states (standing and walking at slow, normal, and fast speeds). Outcome measures included neck flexion, upper-trunk angle, lumbosacral angle, and normalized surface electromyography of the cervical erector spinae (CES) and upper trapezius (UTZ). A three-way ANOVA was used to evaluate main and interaction effects. **Results:** Increasing backpack load significantly increased neck flexion and upper-trunk angle while reducing the lumbosacral angle (all *p* < 0.001). Muscle activity rose proportionally with load, with UTZ activation nearly doubling from 10.7% to 21.1% maximum voluntary contraction (MVC) at 10% body weight. Faster walking increased lumbar flexion and elevated CES and UTZ activation (*p* < 0.05), while neck and upper-trunk postures remained stable across speeds. Females maintained more upright postures but exhibited higher muscle activity than males (*p* < 0.01). UTZ activation frequently exceeded the 15% MVC fatigue threshold during walking with backpacks. **Conclusions:** Walking speed critically modulates musculoskeletal strain during concurrent smartphone use and load carriage. The combined effects of backpack load and smartphone use can elevate neck–shoulder muscle demands, with UTZ activity occasionally approaching fatigue thresholds under moderate load and faster walking. Based on the current findings, backpack loads above 5% of body weight may increase the risk of neck–shoulder strain. Additionally, reducing smartphone use during fast walking may help prevent neck–shoulder fatigue and related musculoskeletal discomfort.

## 1. Introduction

The rapid adoption of smartphones has reshaped how people interact with their environments, with mobile tasks increasingly performed while moving. Recent global statistics indicate that smartphone ownership has reached 89% of the world’s population by early 2025, reflecting the near-universal prevalence of mobile device use across regions [1]. In Taiwan, specifically, 95% of internet users access content primarily through mobile devices [2], further emphasizing the central role of smartphones in daily activities. While smartphones have enhanced connectivity, their frequent and prolonged use has raised ergonomic concerns. Extensive research has shown that sustained device use is associated with an increased risk of musculoskeletal disorders (MSDs), particularly in the neck and shoulder regions, due to forward head posture and elevated cervical loading [3,4,5,6,7].

Importantly, smartphone use is no longer confined to sedentary or standing positions. A large proportion of users now interact with their devices while walking. Surveys indicate that nearly one-third of U.S. users and close to 40% of Taiwanese users habitually engage in non-voice smartphone activities during ambulation [8,9]. This dual-task behavior introduces ergonomic challenges because attention must be shared between visual interaction with the device and the demands of locomotor stability. Studies have shown that smartphone use during walking can alter head and neck alignment, increase cervical flexion, and produce a more pronounced kyphotic curvature than during standing [10,11]. Broader reviews have confirmed these risks, noting that smartphone-induced postural alterations may lead to spinal deformities, reduced gait stability, and increased muscle fatigue [12]. Furthermore, longer usage duration has been associated with reduced neck muscle endurance and more severe neck pain among university students [13,14]. Excessive screen exposure has also been linked to visual strain and disrupted postural control [15], compounding musculoskeletal risks.

Alongside smartphone use, backpack carriage remains a daily necessity for students and young professionals. Observational surveys report that more than 80% of university students use backpacks, often exceeding recommended weight limits [16,17]. Loads above 10% body weight (BW) are particularly concerning, as they are known to induce forward head tilt, posterior trunk lean, and elevated cervical erector spinae (CES) and upper trapezius (UTZ) activity [18,19,20]. These biomechanical adjustments are compensatory in nature but increase the likelihood of discomfort, fatigue, and MSDs development [21,22,23]. Over prolonged periods, such loads may contribute to spinal misalignment and chronic pain [24].

Although the independent effects of smartphone use and backpack carriage have been investigated extensively, their simultaneous impact under dynamic walking conditions has received less attention. prior work demonstrates that smartphone use while walking reduces step length, slows pace, and increases head and neck acceleration compared with sitting or standing [11,25,26]. Other evidence shows that smartphone use while walking can also impair postural balance, as attentional resources are diverted to the screen [27]. Similarly, research on load carriage shows that gait adaptations are speed dependent: as walking speed increases with loads, transverse pelvic and thoracic rotations decrease, stride length shortens, and stride frequency rises [28]. Together, these findings suggest that locomotion speed may be a critical moderator when individuals use smartphones and carry backpacks simultaneously. At faster speeds, biomechanical demands on the cervical and shoulder regions could be magnified, leading to compounded strain that surpasses the effects of each factor in isolation [29,30].

A recent study by our group first examined the combined effects of smartphone use and backpack carriage on posture and neck–shoulder muscle activity in young adults [31]. That work demonstrated that simultaneous device use and load carriage produced compounded strain, with upper trapezius activity often approaching or exceeding the 15% maximum voluntary contraction (MVC) fatigue threshold [32]. However, the analysis was restricted to static standing and self-selected normal walking and therefore did not evaluate how locomotor pace may influence these responses. Because walking speed substantially alters trunk coordination, gait mechanics, and muscular loading during load carriage [28,29,30], it represents a critical but previously unexamined factor in dual-task smartphone and backpack use. Accordingly, the present study extends this prior work by systematically investigating how different walking speeds modulate postural and neuromuscular responses during smartphone use with backpack loads.

Gender differences add further complexity to this issue. Studies indicate that females often display higher relative muscle activation and report greater discomfort than males during smartphone use, despite showing less forward head posture [31,33,34]. Possible explanations include differences in cervical muscle strength, anthropometric characteristics, and strategies for postural stabilization [35,36,37]. In particular, females may experience higher relative CES and UTZ activation to maintain upright postures, which could increase fatigue susceptibility during dual-task conditions [6,38,39,40]. Given the prevalence of smartphone use and backpack carrying in student populations, understanding these gender-specific responses is important for developing tailored ergonomic guidelines.

Taken together, these lines of evidence underscore a critical research gap. While prior studies have examined smartphone use during walking or backpack carriage independently, little is known about how walking speed shapes the combined biomechanical effects of these two common daily activities. Locomotor speed is likely to influence the magnitude of neck–shoulder muscle activation and spinal posture adjustments required to maintain both visual focus on the device and balance under load. Without examining this speed-dependent modulation, ergonomic guidelines may underestimate risks associated with multitasking behaviors typical of modern students and young professionals.

Despite growing evidence that smartphone use and backpack carriage each impose biomechanical demands, no studies have examined how locomotor speed influences these combined effects during walking. Given that speed is a primary determinant of gait mechanics and muscular loading, understanding its role is essential for characterizing real-world dual-task behavior. Therefore, the present study aimed to systematically assess how different walking speeds (slow, normal, fast) modulate postural alignment and neck–shoulder muscle activity during smartphone use with varying backpack loads. By extending our previous work beyond standing and normal walking, this study addresses a critical gap and provides new insight into the additive nature of device use, load carriage, and locomotor demands in young adults.

## 2. Materials and Methods

This study was reviewed and approved by the Ethics Committee of National Taiwan University, Taiwan (Approval code: 202312-EM-051) and conducted in accordance with the principles of the 2013 World Medical Association Declaration of Helsinki. Written informed consent was obtained from all participants, including agreement for the use of anonymized data and images.

A priori power analysis was performed with G*Power (version 3.1.9.7) to estimate the required sample size for a mixed factorial design consisting of one between-subject factor (gender) and two within-subject factors (backpack load and walking speed). Parameters included a medium effect size (f = 0.25), α = 0.05, power = 0.8, moderate correlation among repeated measures (0.5), and an adjusted sphericity correction (ε = 0.75). The analysis indicated that 24 participants (12 per gender) were sufficient to detect medium-sized effects. To enhance power and reduce the risk of data loss, 30 participants were ultimately recruited.

### 2.1. Participants

Thirty university students (15 males and 15 females) participated. Eligibility criteria included: (1) age 18–26 years, (2) daily smartphone use of at least one hour, (3) routine backpack use, and (4) no history of musculoskeletal disorders, injuries, or surgeries in the neck, shoulder, or back within the past year. Participants were excluded if they exhibited structural or functional postural abnormalities that could affect cervical or trunk alignment, including clinically diagnosed forward head posture, scoliosis, abnormal shoulder symmetry, or any neck range of motion limitations exceeding ± 2 standard deviation of normative values [35].

### 2.2. Postural Measurement

Postural variables were recorded using a MacReflex motion analysis system (Qualisys, Gothenburg, Sweden). A digital camera was positioned perpendicular to the sagittal plane to capture movements. Reflective markers were affixed to the tragus (T), shoulder (acromial shelf, S), hip (greater trochanter, H), seventh cervical vertebra (C7), and seventh thoracic vertebra (T7), as shown in Figure 1. To ensure visibility of T7 while wearing a backpack, a 25 cm plastic rod was mounted perpendicular to the sagittal plane, with a marker placed at its distal end. The following angles were computed:-Neck flexion (NF): angle between the C7–tragus line and the tragus–canthus line.-Upper trunk angle (UTA): angle between a horizontal line and the C7–T7 line.-Lumbosacral angle (LSA): calculated using previously established regression equations for L1 and S1 based on trunk angle (TA), which is the flexion angle from the upright position [20,41], as shown in Figure 1.L1 = 1.04 × TA − 1.16        (R^2^ = 0.969)S1 = 34.65 × 1.26 (TA/30)     (R^2^ = 0.916)

Images were recorded during the final 30 s of each trial. Seven frames were extracted at 5 s intervals and analyzed with Qualisys Track Manager (QTM v2025.1). Average values across frames were used for analysis.

### 2.3. Surface Electromyography (EMG) Measurement

Surface EMG signals were acquired from the CES and UTZ on the dominant side using a TeleMyo2400 system (Noraxon, Scottsdale, AZ, USA). Pairs of Ag/AgCl electrodes (1 cm diameter, 2 cm spacing) were positioned following SENIAM guidelines [42] and validated cervical muscle protocols [43,44,45,46,47]. For the CES, electrodes were positioned ~2 cm lateral to the C4 spinous process. For the UTZ, electrodes were placed at the midpoint between the C7 spinous process and acromion.

MVCs were obtained for normalization under standardized isometric contractions following validated protocols [48]. CES MVC was elicited via resisted neck extension, and UTZ MVC via resisted shoulder elevation [40,44]. Each MVC was held for 3–5 s with three repetitions per muscle, separated by ≥30 s rest and a 5 min break between muscles. EMG data were sampled at 1200 Hz, band-pass filtered (20–600 Hz), rectified, and processed. A 0.5 s moving average was applied to determine maximum amplitude, and integrated EMG values were normalized to MVC (% MVC). Recordings were synchronized with posture measurements and obtained during the last 30 s of each trial.

### 2.4. Experimental Design and Procedure

This study employed a mixed factorial design with one between-subject factor (gender: male vs. female) and two within-subject factors: backpack load (0%, 5%, 10% of BW) and locomotor condition (standing and walking at slow, normal, and fast speeds). Smartphone use was included in all conditions, resulting in a total of 12 trials per participant (3 backpack loads × 4 speeds). The order of conditions was randomized to minimize potential order effects.

For the load conditions, participants wore a standardized double-strap backpack (50 × 30 × 18 cm^3^; net weight: 0.2 kg) filled with textbooks to achieve the assigned load. To maintain both experimental control and ecological validity, participants carried the backpack in their typical, symmetrical two-strap manner, with strap adjustments kept natural across individuals. The backpack was positioned over both shoulders at approximately the mid-thoracic level to minimize postural asymmetry, consistent with ergonomic recommendations [21,22]. The selected load magnitudes (0%, 5%, and 10% BW) were based on commonly cited guidelines for safe backpack carriage among young adults, which generally recommend limiting loads to ~10% BW [19,20]. Walking-speed conditions were personalized through a brief treadmill familiarization session (Chanson CS-5728, Taipei, Taiwan), during which each participant identified comfortable slow, normal, and fast walking speeds while using a smartphone. These self-selected paces were then applied consistently throughout the experimental trials to reflect habitual locomotor behavior.

In all conditions, participants performed a standardized smartphone task. Using the LINE messaging application, they continuously exchanged predefined text messages with a research assistant for two minutes [49]. LINE was chosen because messaging represents one of the most common and frequent forms of smartphone interaction in daily life, particularly among young adults. Only this standardized texting task was included in the study; other smartphone activities such as video viewing, gaming, browsing, or navigation—which may impose different visual–postural or muscular demands—were not examined. This task was selected to simulate common device use while ensuring consistency across participants. Each trial lasted two minutes, and postural and muscle activity data were collected during the final 30 s to capture steady-state behavior. To minimize fatigue, participants rested for at least two minutes between trials, with additional breaks provided upon request. On average, the full experimental session required approximately three hours per participant.

### 2.5. Statistical Analysis

All statistical analyses were performed using SPSS 23.0 (IBM Corp., Armonk, NY, USA), with the level of significance set at α = 0.05. Descriptive statistics (mean ± standard deviation) were first calculated for all outcome variables. To examine the effects of experimental conditions, a mixed-design ANOVA was conducted with gender as the between-subject factor and backpack load and walking speed as within-subject factors. The dependent measures included NF, UTA, LSA, and normalized EMG activity of the CES and UTZ. Data distribution was evaluated using the Shapiro–Wilk test for normality, and homogeneity of variance was assessed with Levene’s test. When significant main effects or interactions were identified, Tukey’s HSD tests were applied for post hoc comparisons. Effect sizes were also calculated using power value, with thresholds of 0.2, 0.5, and 0.8 interpreted as small, medium, and large effects, respectively [50]. For the two-level between-subject factor (gender), Cohen’s *d* was also calculated to quantify standardized effect size. Values exceeding 1.0 were considered large according to conventional benchmarks.

## 3. Results

### 3.1. Participant Characteristics and Self-Selected Walking Speeds

Thirty university students (15 males and 15 females) participated in the study. Male participants had a mean age of 21.2 ± 1.6 years, height of 172.6 ± 8.2 cm, and body mass of 67.4 ± 11.5 kg, whereas female participants had a mean age of 21.8 ± 1.5 years, height of 160.3 ± 5.7 cm, and body mass of 52.4 ± 6.3 kg (Table 1). To determine individualized walking speeds, each participant first self-selected their slow, natural, and fast walking paces while using a smartphone without carrying a backpack. The resulting average speeds for males and females are presented in Table 1, with female participants generally selecting lower speeds than their male counterparts.

### 3.2. Three-Way ANOVA Results

The ANOVA revealed significant main effects of backpack weight, walking speed, and gender across most postural and muscle activation variables (Table 2). Backpack weight significantly influenced all dependent measures, including UTA (*p* < 0.05), NF (*p* < 0.001), LSA (*p* < 0.001), CES activity (*p* < 0.001), and UTZ activity (*p* < 0.001). Walking speed also had a marked effect, producing significant changes in LSA (*p* < 0.001), CES activity (*p* < 0.05), and UTZ activity (*p* < 0.01), although no significant differences were observed for UTA (*p* = 0.365) or NF (*p* = 0.899). Gender showed significant effects on UTA (*p* < 0.001), NF (*p* < 0.001), CES activity (*p* < 0.01), and UTZ activity (*p* < 0.01), but not on LSA (*p* = 0.936).

In contrast to these consistent main effects, none of the two-way or three-way interactions among backpack weight, walking speed, and gender reached significance, indicating that the observed effects of load, speed, and gender operated independently rather than synergistically. These results establish that each factor exerted distinct contributions to postural alignment and muscular demand, with backpack weight exerting the broadest influence across variables, followed by gender and walking speed.

### 3.3. Gender Effects

Large gender-related effects were observed, with Cohen’s *d* values exceeding 1.0 for several variables. These results indicate substantial biomechanical differences between males and females despite comparable walking conditions. Significant gender differences were observed in both postural alignment and muscle activation (Figure 2). Males exhibited larger UTA and greater NF compared with females (both *p* < 0.001), indicating a more inclined head and trunk posture during smartphone use with backpack carriage. In contrast, females maintained a more upright posture yet showed significantly higher muscle activation. Across all experimental conditions, CES activity was markedly greater in females than in males (*p* < 0.001), and UTZ activity was also higher (*p* < 0.01). This paradoxical pattern suggests that females require greater relative muscular effort to sustain a posture that is biomechanically more upright, which may increase their susceptibility to fatigue and discomfort under prolonged exposure.

### 3.4. Backpack Weight Effects

Post hoc comparisons demonstrated clear dose–response effects of backpack load on both posture and muscle activity (Table 3). NF increased progressively from 16.7° ± 5.2° without a backpack to 17.6° ± 5.8° at 5% BW and 20.2° ± 6.3° at 10% BW (*p* < 0.001), while UTA followed a similar trend, rising from 26.3° ± 7.8° (no load) to 28.2° ± 6.6° (5% BW) and 28.5° ± 7.5° (10% BW) (*p* < 0.05). In contrast, LSA decreased significantly with increasing load, from 39.4° ± 2.2° at baseline to 35.8° ± 2.1° at 5% BW and 35.5° ± 2.1° at 10% BW (*p* < 0.001), reflecting greater lumbar flexion under load.

Muscle activation showed even stronger load-dependent changes. CES activity increased from 13.6% ± 6.3% MVC with no load to 15.3% ± 5.1% MVC at 5% BW and 17.6% ± 5.7% MVC at 10% BW (*p* < 0.001). The UTZ exhibited the most pronounced effect, with activity nearly doubling from 10.7% ± 5.3% MVC without a backpack to 15.6% ± 5.2% MVC at 5% BW and 21.1% ± 5.3% MVC at 10% BW (*p* < 0.001). These findings indicate that although neck and upper-trunk postures remained relatively stable across walking speeds, lumbar flexion increased as participants walked faster. In contrast, the elevated neck–shoulder muscle activation observed at higher speeds was driven primarily by backpack load rather than by changes in head or trunk posture.

### 3.5. Speed Effects

Walking speed produced distinct effects on spinal posture and muscle activation (Table 4). For postural measures, only the LSA was significantly influenced by speed. The highest value was observed during standing (38.9° ± 2.1°), followed by slow and normal walking (both 37.2° ± ~2.0°), and the lowest during fast walking (36.5° ± 2.2°) (*p* < 0.001), indicating a slight but progressive increase in lumbar flexion (i.e., reduced LSA) with increasing walking speed. In contrast, neither NF nor UTA differed significantly across speeds, suggesting that participants prioritized maintaining a stable head–neck orientation for smartphone viewing regardless of locomotor pace.

In terms of muscle activity, both the CES and UTZ demonstrated speed-related increases. CES activity rose gradually from 13.5% ± 6.1% MVC while standing to 14.9% ± 5.9% MVC during slow walking, 15.8% ± 5.7% MVC during normal walking, and 16.8% ± 5.8% MVC during fast walking (*p* < 0.05). A similar trend was observed for the UTZ, with activity increasing from 14.7% ± 6.7% MVC in standing to 15.3% ± 6.7% MVC in slow walking, 15.8% ± 6.5% MVC in normal walking, and peaking at 17.5% ± 6.8% MVC in fast walking (*p* < 0.01). These findings indicate that faster walking amplifies muscular demands in the neck–shoulder region, even though head and trunk postures remain relatively stable across speeds.

### 3.6. Combined Effects and Fatigue Considerations

Although no significant interaction effects were detected among backpack load, walking speed, and gender, the combined exposure patterns revealed notable ergonomic risks. As shown in Figure 3, both CES and UTZ muscle activity increased progressively with higher loads and faster walking speeds, with the trapezius displaying sharper rises than the erector spinae. Importantly, UTZ activation frequently exceeded the fatigue threshold of 15% MVC under dual-task conditions. Even with a 5% BW backpack, mean UTZ values surpassed this threshold during walking, and further increases were observed at 10% BW across all speeds. CES activity also rose with load and speed, but its values generally remained closer to the threshold, suggesting a lower relative fatigue risk compared with UTZ.

These findings underscore that, even without significant statistical interactions, the simultaneous demands of smartphone use, backpack carriage, and locomotor speed impose compounded strain on the cervical–shoulder complex. The trapezius muscle, in particular, appears highly sensitive to incremental increases in both load and pace, making it a critical site of potential fatigue and discomfort during prolonged daily activities such as commuting or campus walking.

## 4. Discussion

This study examined how backpack load, walking speed, and gender influence postural alignment and neck–shoulder muscle activity during smartphone use. The results demonstrated that all three factors exerted significant and independent effects, with backpack load showing the broadest influence on both posture and muscle activation, walking speed primarily modulating lumbar posture and muscular demand, and gender shaping alignment strategies and relative activation levels. Although no significant interactions were detected, the combined exposure patterns highlighted important ergonomic risks, particularly for the trapezius muscle, which frequently exceeded the fatigue threshold of 15% MVC [32] under conditions that are common in students’ daily routines. These findings not only confirm earlier reports on the risks of smartphone use and backpack carriage but also extend the literature by revealing how walking speed magnifies musculoskeletal demands, thereby identifying locomotor pace as a critical moderator of ergonomic risk.

### 4.1. Postural Adaptations and Biomechanical Mechanisms

The present findings confirmed that smartphone use with backpack carriage induces distinct postural adaptations and muscular demands. Consistent with earlier studies, participants increased NF and trunk inclination while reducing lumbar curvature under load [16,18,19,20,21,22,23]. These adjustments reflect biomechanical strategies to counterbalance the posterior shift in center of gravity from backpack load while maintaining visual attention on the smartphone. However, such compensations come at the cost of elevated activity in the CES and UTZ. The trapezius, in particular, demonstrated pronounced increases, corroborating evidence that this muscle is highly sensitive to sustained postural demands and load stabilization [24].

Interestingly, head–neck orientation remained relatively stable across speeds, suggesting that participants prioritized visual alignment with the smartphone. This aligns with reports that smartphone use imposes strong visual constraints, often overriding natural locomotor adjustments [10,11]. Thus, the observed postural changes can be understood as a redistribution of effort across spinal segments, with lumbar adjustments and elevated neck–shoulder muscle activity compensating for the fixed head–neck posture required by the task.

### 4.2. Speed Effects

Whereas our earlier work identified compounded strain under dual-task conditions [31], the present study extends these findings by demonstrating that walking speed further amplifies muscular demands, particularly in the trapezius. A central contribution of this study lies in clarifying how walking speed modulates these biomechanical responses. Faster walking was associated with greater lumbar flexion (reduced LSA) and progressively higher CES and UTZ activation, while NF and UTA were unaffected. This suggests that participants constrained head and trunk orientation to maintain stable visual focus on the smartphone but compensated through increased lumbar flexion and muscular effort.

Previous work on smartphone use during walking has shown altered gait kinematics, reduced step length, and increased head acceleration [10,11,25,26]. Studies of load carriage similarly report that faster walking reduces trunk rotation and increases stride frequency, thereby elevating musculoskeletal loading [28,29,30]. The present findings bridge these studies by showing that speed magnifies muscular demands in dual-task contexts, with UTZ activation consistently exceeding the 15% MVC fatigue threshold at moderate loads during walking. This highlights walking speed as a critical moderator of ergonomic risk, underscoring that seemingly routine behaviors—such as texting while briskly walking with a backpack—can create sustained muscular demands sufficient to induce fatigue and discomfort.

Although the walking speeds in this study were self-selected, the mean values for the slow, normal, and fast conditions (Table 1) fall within the ranges commonly reported in smartphone-related gait studies, where dual-task demands typically reduce preferred walking speed. It is therefore expected that our speeds are lower than normative values for adults walking without secondary tasks, such as those reported by Bohannon [51], who documented comfortable walking speeds of approximately 5 km/h in healthy adults. Similarly to our earlier work, which observed limited changes in cervical posture across self-selected speeds during smartphone use [11], the present findings suggest that locomotor pace itself may play a smaller role than the dynamic nature of walking in shaping head and upper-trunk behavior. Studies by Yoon et al. [26] and Han and Shin [52], which also utilized preferred or self-selected speeds, reported modest variations in neck and trunk posture across gait conditions, consistent with our results. Thus, while reporting actual walking speeds enhances reproducibility, the relative differences among the three self-selected speed levels were sufficient for capturing speed-related biomechanical trends during smartphone use.

### 4.3. Gender Differences

Gender-based differences were evident in both posture and muscular demands. Males demonstrated greater NF and UTA, adopting a more forward-inclined posture, whereas females maintained more upright alignment. However, females exhibited significantly higher CES and UTZ activity, a paradoxical pattern consistent with prior reports that females often display greater muscle activation and discomfort despite less forward flexion [6,33,34,35,36,37,38,40].

Despite maintaining a more upright neck and upper-trunk posture, female participants consistently exhibited higher CES and UTZ activation than males. This pattern indicates that a visually better posture does not necessarily translate to reduced muscular demand. Prior work shows gender differences in smartphone behaviors and neck posture [36] and reports greater neck–shoulder strain among female smartphone users under similar tasks [6]. In addition, device exposure is associated with reduced neck muscle endurance and heightened fatigue/pain responses [13,14], implying higher relative effort for females given typically lower absolute neck–shoulder strength. At a broader level, systematic evidence links touchscreen use to musculoskeletal symptoms and exposure profiles that can elevate cervical loading [15], and technology-use tasks demonstrate sensitivity of superficial neck–shoulder muscles (including UTZ and CES) to stabilization demands [43]. Together, these factors help explain why females showed more upright alignment yet experienced greater neck–shoulder muscle activation.

These gender differences may stem from disparities in cervical muscle strength, anthropometric characteristics, and stabilization strategies [39]. Females generally have lower neck–shoulder muscle strength and smaller cervical musculature, increasing the relative effort required to maintain upright posture during multitasking. This higher relative activation was consistent across backpack loads and locomotor conditions, suggesting a general sex-specific response. Given their greater risk of neck–shoulder discomfort [12,13,15], the added demands of smartphone use while carrying a backpack may further heighten vulnerability in females.

### 4.4. Combined Effects

Although backpack load, walking speed, and gender each exerted significant but independent effects on postural alignment and muscle activation, no significant two-way or three-way interactions were detected. This indicates that these factors affected musculoskeletal responses through separate pathways rather than synergistic mechanisms. In practical terms; however, their impacts can accumulate during real-world multitasking. The compounded strain described in this study therefore reflects the additive effects of these independent demands rather than statistically significant interactions. Heavier backpack loads increased cervical and upper-trunk flexion and substantially elevated CES and UTZ activation, whereas faster walking increased lumbar flexion and further raised muscle activity. When these independent effects occur simultaneously—such as walking quickly while carrying a backpack and using a smartphone—the overall musculoskeletal burden increases cumulatively, even in the absence of statistical interactions. This interpretation reconciles the practical implications with the statistical findings and reflects the real-world experience of concurrent biomechanical stressors.

CES activity also increased with load and speed, though its values generally remained closer to the 15% MVC fatigue threshold, suggesting a lower relative fatigue risk compared with UTZ. In contrast, UTZ activation frequently approached or exceeded this threshold, particularly under moderate loads and faster walking speeds. Although UTZ activation occasionally reached or slightly exceeded the classical 15% MVC fatigue threshold at the 5% BW backpack load, the average UTZ activation at this load (15.6 ± 5.2% MVC; Table 3) suggests that the threshold crossings likely reflected intermittent increases associated with the dual-task demands of smartphone use rather than prolonged static loading. In contrast, the 10% BW load produced substantially higher UTZ activation (21.1 ± 5.3% MVC), indicating a clearer elevation in fatigue risk.

It should be noted that although Rohmert’s classical 15% MVC fatigue threshold remains a widely used benchmark in ergonomics, subsequent fatigue research and endurance-time studies have expanded our understanding of muscle loading and local fatigue [53,54,55]. For example, Bystrom and Fransson-Hall [54] demonstrated that the acceptability of sustained or intermittent contractions varies with contraction intensity, while Frey Law and Avin [55] reported that endurance time is joint-specific rather than governed by a single generalized fatigue curve. These later findings do not alter the interpretation of the present results but provide additional context for understanding fatigue-related risk, particularly given the elevated UTZ activation observed under combined load and speed conditions.

### 4.5. Ergonomic Implications

From an applied perspective, these findings provide guidance for risk reduction in everyday student activities. First, although these exceedances were modest and speed-dependent, the findings suggest that backpack loads exceeding 5% BW may increase neck–shoulder strain during smartphone use while walking. Second, individuals should be advised to minimize smartphone use while walking at faster speeds, since speed amplifies muscular demands even when head–neck orientation remains stable. Educational campaigns in schools and universities could emphasize safe load limits, the importance of posture awareness, and strategies to reduce smartphone multitasking while walking.

In addition, ergonomic design solutions may help mitigate risks. Lightweight backpacks, improved load distribution systems, or smartphone interfaces optimized for quick, minimal interactions during walking could reduce musculoskeletal strain. Public health strategies could further highlight the cumulative risks of sustained dual-task behaviors, especially among students who frequently engage in smartphone use during commuting.

### 4.6. Limitations and Future Directions

Several limitations should be considered when interpreting these findings. First, this study used treadmill walking, which ensured controlled experimental conditions but may not fully reflect the variability and natural adjustments of overground walking in real-world environments. Second, a no-phone walking condition was not included, limiting the ability to quantify the incremental biomechanical impact of smartphone use relative to natural gait. Although the present study focused on how backpack load and walking speed influence posture and muscle activity during smartphone interaction, future research should incorporate both phone-use and no-phone trials to better isolate the specific contribution of smartphone engagement. Third, only a single smartphone task (texting) was examined; other common activities such as gaming, browsing, or navigation may impose different visual–postural or muscular demands and warrant further investigation. Fourth, the trial duration was limited to two minutes, which likely underestimates the cumulative effects of prolonged smartphone use and backpack carriage typical in daily life. This short exposure also prevented evaluation of potential time-dependent postural drift or fatigue-related changes within each trial, as beginning–end posture comparisons were not extracted under the current protocol. Future studies should therefore adopt longer durations and time-segmented analyses to better capture progressive posture adaptations as fatigue develops. Fifth, the sample comprised healthy young adults (university students), restricting generalizability to older individuals or populations with musculoskeletal vulnerabilities. Sixth, although this study analyzed postural and EMG outcomes, complementary gait parameters such as stride length, cadence, and stability measures were not assessed and could provide additional insight into locomotor adaptations.

Beyond these methodological considerations, this study is the first to demonstrate that walking speed amplifies musculoskeletal strain under smartphone–backpack conditions; only three self-selected speed levels were tested. Future research should explore a wider spectrum of locomotor paces, including real-world variations in step dynamics and terrain, to refine ergonomic guidelines. Incorporating longer exposure times, diverse populations, and multiple smartphone tasks will further strengthen the ecological validity of this line of research.

## 5. Conclusions

This study examined how backpack load and self-selected walking speed influence posture and neck–shoulder muscle activation during smartphone use. Heavier backpack loads increased cervical and upper-trunk flexion and elevated CES and UTZ activation, whereas faster walking primarily increased lumbar flexion and muscular demands while leaving neck and upper-trunk angles relatively unchanged. Although no significant interactions were found among load, speed, and gender, the combined influence of these independent factors resulted in cumulative musculoskeletal strain, especially reflected in the higher UTZ activation across conditions. Backpack loads above 5% BW may elevate UTZ activation and contribute to increased neck–shoulder strain, particularly when combined with faster walking. The self-selected speed values reported in this study also provide a useful reference for future work examining gait-related biomechanical responses during mobile device interaction.

Overall, the findings highlight the additive nature of loading and gait demands during everyday smartphone use and underscore the importance of considering multitasking contexts in ergonomic evaluations. Future research incorporating no-phone control conditions, longer exposure durations, diverse smartphone tasks, and overground walking environments will further strengthen the ecological relevance and generalizability of this line of research.

## Figures and Tables

**Figure 1 healthcare-13-03141-f001:**
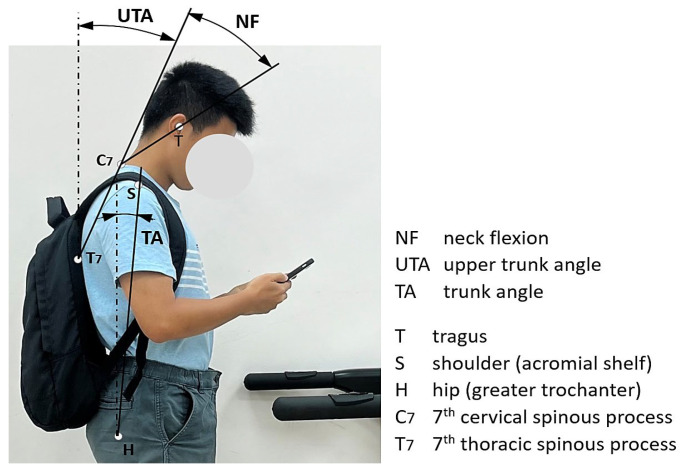
Schematic representation of marker placements and angle definitions for postural measurements, with the lumbosacral angle derived from the trunk angle.

**Figure 2 healthcare-13-03141-f002:**
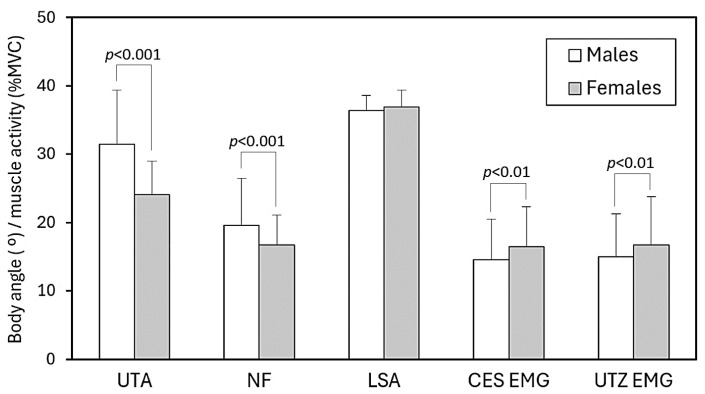
Main effects of measured angles and muscle activities in male and female participants (UTA: upper trunk angle; NF: neck flexion; LSA: lumbosacral angle; CES EMG: cervical erector spinae activity; UTZ EMG: upper trapezius activity; MVC: maximum voluntary contraction).

**Figure 3 healthcare-13-03141-f003:**
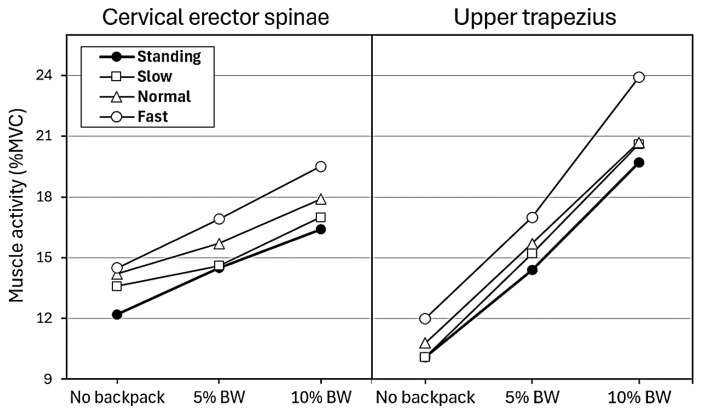
Cervical erector spinae and upper trapezius activities across backpack load conditions during standing and three walking speeds (slow, normal, and fast). MVC, maximum voluntary contraction; BW, body weight.

**Table 1 healthcare-13-03141-t001:** Participant demographics (mean ± SD) and self-selected slow, normal, and fast walking speeds for male and female participants.

Items	Men (*n* = 15)	Women (*n* = 15)
Age (years)	21.2 (1.6)	21.8 (1.5)
Height (cm)	172.6 (8.2)	160.3 (5.7)
Body mass (kg)	67.4 (11.5)	52.4 (6.3)
Self-selected walking speeds (km/h)		
Slow	2.3 (0.5)	2.2 (0.4)
Normal	3.9 (0.6)	3.7 (0.6)
Fast	4.8 (0.8)	4.5 (0.6)

**Table 2 healthcare-13-03141-t002:** Three-way ANOVA results for postural alignment and neck–shoulder muscle activity across backpack loads, walking speeds, and gender.

Variables	Responses	F	*p*	Power	Cohen’s *d*
Backpack weight	Upper trunk angle	4.02	<0.05	0.715	—
	Neck flexion	11.77	<0.001	0.994	—
	Lumbosacral angle	32.78	<0.001	1.000	—
	Cervical erector spinae	14.47	<0.001	0.999	—
	Upper trapezius	119.15	<0.001	1.000	—
Speed	Upper trunk angle	1.06	0.365	0.287	—
	Neck flexion	0.20	0.899	0.087	—
	Lumbosacral angle	11.93	<0.001	1.000	—
	Cervical erector spinae	2.68	<0.05	0.651	—
	Upper trapezius	4.54	<0.01	0.883	—
Gender	Upper trunk angle	45.16	<0.001	1.000	2.45
	Neck flexion	28.71	<0.001	0.996	1.96
	Lumbosacral angle	0.01	0.936	0.051	0.04
	Cervical erector spinae	10.08	<0.01	0.886	1.16
	Upper trapezius	7.61	<0.01	0.785	1.01
Backpack weight × Speed	Upper trunk angle	0.26	0.953	0.122	—
	Neck flexion	0.41	0.873	0.170	—
	Lumbosacral angle	0.26	0.954	0.121	—
	Cervical erector spinae	0.32	0.927	0.139	—
	Upper trapezius	0.20	0.978	0.101	—
Backpack weight × Gender	Upper trunk angle	0.08	0.922	0.062	—
	Neck flexion	0.50	0.606	0.132	—
	Lumbosacral angle	0.35	0.703	0.107	—
	Cervical erector spinae	0.01	0.990	0.052	—
	Upper trapezius	0.10	0.907	0.065	—
Speed × Gender	Upper trunk angle	0.23	0.878	0.092	—
	Neck flexion	0.08	0.971	0.064	—
	Lumbosacral angle	1.18	0.319	0.315	—
	Cervical erector spinae	0.31	0.819	0.110	—
	Upper trapezius	0.44	0.723	0.139	—
Backpack weight × Speed × Gender	Upper trunk angle	0.29	0.941	0.131	—
	Neck flexion	0.10	0.996	0.074	—
	Lumbosacral angle	0.47	0.828	0.193	—
	Cervical erector spinae	0.08	0.998	0.069	—
	Upper trapezius	0.07	0.999	0.067	—

Note: Cohen’s *d* is reported only for the two-level between-subject factor (Gender).

**Table 3 healthcare-13-03141-t003:** Post hoc Tukey comparisons of postural alignment and neck–shoulder muscle activity across different backpack load conditions.

Backpack Loads	UTA (°)	NF (°)	LSA (°)	CES (%MVC)	UTZ (%MVC)
No backpack	26.3 (7.8) A	16.7 (5.2) A	39.4 (2.2) A	13.6 (6.3) A	10.7 (5.3) A
5% of body weight	28.2 (6.6) B	17.6 (5.8) A	35.8 (2.1) B	15.3 (5.1) B	15.6 (5.2) B
10% of body weight	28.5 (7.5) B	20.2 (6.3) B	35.5 (2.1) B	17.6 (5.7) C	21.1 (5.3) C

Note: Data (mean, with standard deviation in parentheses) with the same letter do not differ in the Tukey test. UTA, upper trunk angle; NF, neck flexion; LSA, lumbosacral angle; CES, cervical erector spinae; UTZ, upper trapezius; MVC, maximum voluntary contraction.

**Table 4 healthcare-13-03141-t004:** Post hoc Tukey comparisons of postural alignment and neck–shoulder muscle activity across speed conditions (standing and walking at slow, normal, and fast speeds).

Speeds	UTA (°)	NF (°)	LSA (°)	CES (%MVC)	UTZ (%MVC)
Standing	26.7 (6.8) A	18.2 (6.5) A	38.9 (2.1) A	13.5 (6.1) A	14.7 (6.7) A
Slow	28.3 (7.2) A	18.5 (6.1) A	37.2 (2.1) B	14.9 (5.9) AB	15.3 (6.7) A
Normal	27.8 (7.9) A	17.9 (5.5) A	37.2 (2.0) B	15.8 (5.7) AB	15.8 (6.5) A
Fast	27.9 (7.6) A	17.9 (5.8) A	36.5 (2.2) B	16.8 (5.8) B	17.5 (6.8) B

Note: Data (mean, with standard deviation in parentheses) with the same letter do not differ in the Tukey test. UTA, upper trunk angle; NF, neck flexion; LSA, lumbosacral angle; CES, cervical erector spinae; UTZ, upper trapezius; MVC, maximum voluntary contraction.

## Data Availability

The data presented in this study are available on request from the corresponding author. The data are not publicly available due to privacy and ethical restrictions.

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
