# Peer review of "Walking Speed Modulates Neck–Shoulder Strain During Smartphone Use with Backpack Load"

_healthcare, 2025, doi:10.3390/healthcare13233141_

Round 1

Reviewer 1 Report

Comments and Suggestions for Authors

I would like to thank the authors for their contribution to the understanding of posture and muscle activity under varying walking conditions when using smartphones with backpacks. The topic is relevant, and the experimental approach is interesting. However, several aspects require clarification and revision to improve the interpretability of the manuscript.

  1. Please review the manuscript for English errors. For instance, line 81 contains a typographical error (“fist”).
  2. Ensure consistent terminology or define abbreviations clearly upon first use. Example: “CES and UTZ activity” should consistently correspond to “cervical erector spinae and upper trapezius activity.”
  3. Reference [1] reports smartphone statistics specific to Taiwan, but the statement generalizes the statistics to all of the developed countries. Please consider adding more recent and globally representative data such as that from SQ Magazine (2025) : “Smartphone ownership has reached an unprecedented 89% of the global population by early 2025” (https://sqmagazine.co.uk/smartphone-statistics/).
  4. The table title should be more specific to the study context and variables presented.
  5. Line 252 appears to be missing a “±” symbol for variability reporting (e.g., mean ± SD).
  6. Lines 289–290 state :
    “These findings indicate that faster walking amplifies muscular demands in the neck–shoulder region, even though head and trunk postures remain relatively stable across speeds.”
    However, line 273 and the abstract mention “greater lumbar flexion as walking speed increased.” These statements could appear contradictory. Please clarify that the neck posture remains stable while lumbar flexion increases with speed and specify whether load or speed primarily drives neck–shoulder activation.
  7. The gender-related findings "females showing more upright posture but higher muscle activation" warrant further discussion. Why does a seemingly better posture result in greater strain? Possible explanations might include differences in muscle strength, biomechanical alignment, or compensatory activation mechanisms.
  8. The results indicate no significant interactions between walking speed, load, and gender (lines 293–295), yet the discussion refers to compounded strain when these factors are combined. This disconnect should be addressed to clarify whether the practical observations suggest trends that were not statistically significant or if this refers to cumulative effects across independent factors.
  9. The manuscript states that even a 5% backpack load exceeded the fatigue threshold for UTZ activity during walking (line 382) while simultaneously recommending keeping loads below 5%. This appears inconsistent unless the threshold exceedance was marginal or context-specific. Please clarify whether “exceeded” refers to transient peaks or sustained activation and whether the recommendation is based on safety margins or conservative practice.

Reviewer 2 Report

Comments and Suggestions for Authors

The title of this paper really caught my attention "

"Walking Speed Modulates Neck–Shoulder Strain during Smartphone Use with Backpack Load"

I assumed that this would examine the effects of smart phone use and backpacks on posture, walking, and fatigue. This was probably my mistake. All of the trials used the same smart phone task, so it is impossible to infer anything about smart phone. I would expect to see at least one trial without the phone. 

This leaves us with a study of just backpack weight and walking speed. The walking speeds were self selected and not reported. The absence of walking speeds makes it hard to really interpret the results. Certainly it would be hard for anyone to reproduce these results in the absence of this information. The authors report significant interactions with walking speeds and other variables, but they are small and the variance of the responses are high -- see Tables 1 and 3.  

It would be interesting to see what kind of differences would be seen for phone use versus no phone use.

The fatigue threshold is based on the classic work of Rohmer. There has been much fatigue research since this work was published in 1973. Some good examples include;

  • EDWARDS R. (1986) Muscle fatigue and pain. Acta Medica Scandinavica, 220(S711), pp.179-188.
  • Byström S, Fransson-Hall C. Acceptability of intermittent handgrip contractions based on physiological response. Human Factors. 1994;36(1):158-71.
  • Frey Law L, Avin K. Endurance time is joint-specific: a modelling and meta-analysis investigation. Ergonomics. 2010;53(1):109-29.

I see the short 2 minute trials as another limitation of this study. I would expect to see some posture changes as subjects fatigue. Any such changes are confounded by the experimental design. As a minimum the postures should be reported at the beginning and end of the each set of trials.

Aside from the limitations regarding the aforementioned concerns, the experimental design and statistical analysis and the writing appear to be very good.

Reviewer 3 Report

Comments and Suggestions for Authors

This study investigated how walking speed, backpack load, and gender affect postural alignment and neck-shoulder muscle activity during smartphone use. The study is quite innovative in terms of what it adds to the literature. 

A few suggestions are as follows:

Abstract: The abstract is well-structured and adequately explains the study's purpose and findings.

Introduction: The rationale for the study is appropriately listed in the introduction, and the hypotheses are clearly stated.

Method: The EMG protocols comply with SENIAM standards and are explained in detail.

"LINE messaging" was selected as the phone task. In real life, various tasks (watching videos, gaming, navigation) can create different muscle loading; this limitation should be clearly stated in the methods section.

Results: Tables and figures are clearly and systematically arranged. Providing effect sizes (Cohen's d) is a good practice.

Discussion

The findings are systematically related to the literature. In the gender differences section, not only muscle activation but also anthropometric differences can be discussed.

Conclusions are presented clearly and in a practical manner.

The statement “should remain below 5% BW” is overly definitive given the study’s data limitations. It would be more appropriate to use a cautious phrasing such as: “Based on the current findings, backpack loads above 5% of body weight may increase the risk of neck–shoulder strain.”This revision better reflects the evidence without overstating the generalizability of the results.

Round 2

Reviewer 2 Report

Comments and Suggestions for Authors

While the authors have provided more details, particularly those of a statistical nature, I I don't find a significant theoretical and practical contribution that merits publications. As I mentioned in my previous review, the study is fundamentally flawed by the experimental design. The conclusions are obvious, e.g., "Heavier backpack loads increased cervical and upper-trunk flexion and elevated CES and UTZ activation ..." and vague: "Backpack loads above 5% BW may elevate UTZ activation and contribute to increased neck–shoulder strain ..." This doesn't really give me new information that informs my understanding of the backpacks or the design and fitting of backpacks. The authors further conclude: "Overall, the findings highlight the additive nature of loading and gait demands during everyday smartphone use and underscore ..." again this provides little usable information.

To the authors credit they acknowledge that have a standing and a no backpack treatment is a limitation, but this is still a limitation of the work. 

As I reflect on this, another serious limitation of this experimental design is the backpack position, i.e., the distance between the backpack center of gravity and the L5/S1 or C7/T1 vertebrate.

 I also want to call the authors attention to a publication by Bohannon (1997) that provides some baseline data for normal walking speeds.

Bohannon, Richard W. Comfortable and maximum walking speed of adults aged 20—79 years: reference values and determinants. Age and Aging, 26.1 (1997): 15-19.

I'm sorry that I can't provide a more favorable review. I hope that I have provided some useful information to you for future work.
